# Tracing the Origin of Cell-Free DNA Molecules through Tissue-Specific Epigenetic Signatures

**DOI:** 10.3390/diagnostics12081834

**Published:** 2022-07-29

**Authors:** Angela Oberhofer, Abel J. Bronkhorst, Carsten Uhlig, Vida Ungerer, Stefan Holdenrieder

**Affiliations:** Munich Biomarker Research Center, Institute of Laboratory Medicine, German Heart Center, Technical University Munich, Lazarettstraße 36, D-80636 Munich, Germany; oberhofer@dhm.mhn.de (A.O.); bronkhorst@dhm.mhn.de (A.J.B.); uhlig@dhm.mhn.de (C.U.); ungerer@dhm.mhn.de (V.U.)

**Keywords:** cell-free DNA, epigenetics, liquid biopsy, tissue-of-origin

## Abstract

All cell and tissue types constantly release DNA fragments into human body fluids by various mechanisms including programmed cell death, accidental cell degradation and active extrusion. Particularly, cell-free DNA (cfDNA) in plasma or serum has been utilized for minimally invasive molecular diagnostics. Disease onset or pathological conditions that lead to increased cell death alter the contribution of different tissues to the total pool of cfDNA. Because cfDNA molecules retain cell-type specific epigenetic features, it is possible to infer tissue-of-origin from epigenetic characteristics. Recent research efforts demonstrated that analysis of, e.g., methylation patterns, nucleosome occupancy, and fragmentomics determined the cell- or tissue-of-origin of individual cfDNA molecules. This novel tissue-of origin-analysis enables to estimate the contributions of different tissues to the total cfDNA pool in body fluids and find tissues with increased cell death (pathologic condition), expanding the portfolio of liquid biopsies towards a wide range of pathologies and early diagnosis. In this review, we summarize the currently available tissue-of-origin approaches and point out the next steps towards clinical implementation.

## 1. Introduction

Liquid biopsy is a minimally invasive diagnostic approach in which a variety of biomarkers present in body fluids (mostly plasma and serum), including nucleic acids, are analyzed in order to detect diseases, e.g., cancer. The analysis of circulating nucleic acids already complements ‘classical’ solid biopsy (i.e., tissue biopsy) by providing a more comprehensive picture of the progression and heterogeneity of cancer, the response of tumors to therapy, the presence of minimal residual disease, and has recently been suggested as promising screening tool for early cancer diagnosis. Liquid biopsy has significantly advanced prenatal testing for genetic disorders and monitoring of graft rejection. Due to its minimally invasiveness, it offers the possibility to obtain serial snapshots during disease progression and is well suited for monitoring of individual therapy response. So far, non-invasive prenatal testing (NIPT) [1], detection of circulating tumor DNA (ctDNA) in the plasma of cancer patients [2], and detection of donor-derived DNA in the plasma of transplantation recipients [3] have been clinically implemented. These analyses are all based on genetic differences (fetal and maternal DNA; donor and recipient DNA in graft patients) or mutations (cancer), limiting the approach to diseases involving genetic aberrations. Additionally, detection of the low number of mutated cfDNA molecules poses a great analytical challenge and lies beyond the current limit of detection at early cancer stages. Thus, screening tests for early detection of many types of cancer and various pathologies are still lacking and are urgently needed. One way to fill this diagnostic gap might be to utilize more general features carried by all cfDNA molecules, such as epigenetic characteristics. This might broaden their diagnostic applications and expand them to early diagnosis of a wide range of pathologies.

DNA fragments from virtually all cell and tissue types are constantly released into various human body fluids, e.g., plasma, serum, cerebrospinal fluid, and urine [4,5,6]. Exogenous DNA (e.g., bacterial or viral DNA) can also be found in the different body fluids [7,8,9,10,11] in addition to host DNA (genomic and mitochondrial DNA). The so-called cell-free DNA (cfDNA) in plasma is very short-lived (half-time between 15 min and 2.5 h [12,13]) and is released into the circulation by different cellular pathways including different sorts of cell death, regular cellular turnover and upon pathologies [11,14,15,16]. Thereby, liquid biopsy provides a current snapshot of what cell or tissue types contribute to the plasma DNA pool at the time of blood draw. In diseased tissues, more cell death has been observed [17,18], and consequently, more cfDNA molecules from that particular cell type(s) are released into the blood compared with healthy individuals. Enzymatic digestion is usually involved in DNA release and, e.g., in plasma, cfDNA predominantly exists as short double-stranded DNA fragments in the size between 100 and 200 base pairs (bp). However, considerably longer fragments (almost 24,000 bp) have been detected recently [19]. It is widely accepted that cfDNA molecules retain the cell-type specific epigenetic features and it has been shown that methylation or fragmentation patterns of cfDNA molecules are cell-type and tissue-specific [20,21,22,23], as genes are differentially regulated in distinct cell types and different release pathways are employed by distinct cell types. Furthermore, fragment length and fragment end motifs of cfDNA molecules are non-random and depend on the cellular release pathway, the involved enzymes, and regulatory state of the releasing tissue [15,24,25,26]. Therefore, methylation or fragmentation patterns of cfDNA molecules represent epigenetic features that are well-applicable to infer tissue-of-origin of individual cfDNA molecules and thereby determine the contribution of individual cell types or tissues to the total plasma cfDNA pool. During the last years, research efforts have focused on identifying cell type-specific methylation or fragmentation patterns that can be employed to trace the origin of individual cfDNA molecules (Figure 1). Various groups developed methods to assign cfDNA molecules to their cell- or tissue-of-origin based on epigenetic features such as DNA methylation patterns, nucleosome footprinting, transcription factor binding sites, fragmentation patterns, and histone modifications among others [20,21,23,27,28,29]. Establishing methylation or fragmentation pattern atlases of distinct cell types is a prerequisite for developing sensitive epigenetic-based tissue-of-origin analyses. Deciphering these patterns and attributing them to the corresponding cell- or tissue-of-origin opens up novel ways for non-invasive early diagnostic tests. The focus on more general cfDNA characteristics holds promise to develop multiple powerful diagnostic tools in the future. In this review, we will discuss the currently available toolbox for liquid biopsies and the early detection of diseases based on the tissue-of-origin analysis.

## 2. Epigenetic-Based Biomarkers in Liquid Biopsy

Gene transcription is a tightly regulated process that involves multiple epigenetic regulatory mechanisms, such as DNA methylation, DNA compaction in accessible and non-accessible chromatin regions through binding of histones, and transcription factor (TF) binding. Transcriptional programs differ significantly between cell types and are altered upon tumorigenesis or onset of pathology. Investigating the differences in these epigenetic patterns in circulating nucleic acids might enable the early diagnosis of a wide range of pathologies. In the following section, we summarize currently developed approaches that infer tissue-of-origin from cfDNA molecules based on different epigenetic characteristics (see Table 1 for an overview of important approaches). Handling of plasma DNA molecules necessitates very high caution with respect to DNA isolation and treatment procedures to account for the very low amounts of available DNA (very low input DNA) and its already fragmented nature. Therefore, existing protocols for the analysis of DNA from, e.g., tissue need to be adapted to the special needs of circulating nucleic acids from blood. We will also include information on cfDNA-adapted protocols within this review.

### 2.1. Methylation-Based Tissue-of-Origin Analysis

The transfer of a methyl group onto the C5 position of cytosine residues results in the formation of 5-methylcytosine (5 mC) and occurs almost exclusively on cytosines with an adjacent guanosine, which are termed CpG sites. This DNA modification inhibits gene expression by hindering the binding of TFs to DNA or by inhibiting the recruitment of proteins involved in gene expression. Thus, methylated promoters are usually a sign of a silenced gene with no or low gene expression. Transcriptional programs are tightly regulated by cell-specific DNA methylation patterns and these methylation marks reflect the gene expression profile of the respective cell type. Methylation patterns are conserved among cells of the same cell type, are highly stable under physiological or pathological conditions, and are unique to each cell type. In cancer cells, transcriptional programs are altered and lead to different methylation patterns. Promoter regions of tumor suppressor genes are generally hypermethylated (i.e., silenced gene expression), whereas promoters of cancer driver genes are hypomethylated (i.e., activated gene expression) [41,42].

There are numerous different techniques to identify methylated cytosines. Generally, it can be differentiated between genome-wide, reduced genome and targeted approaches; the order mentioned represents increasing cost effectiveness. For a complete and detailed description of all available DNA methylation detection approaches, please refer to the recent review by Galardi et al. [43]. In order to detect 5 mCs, the DNA needs to be treated either by (i) restriction enzyme digestion, (ii) bisulfite treatment, (iii) affinity enrichment of methylated DNA or (iv) a combined treatment of enzymatic and chemical modifications. The gold standard for genome-wide methylation analysis is bisulfite conversion [44,45]. For this purpose, unmethylated cytosine residues are converted into uracil residues, while methylated cytosines remain unmodified. During PCR, uracil residues are replaced by thymine residues and subsequent analysis observes cytosine residues as methylated cytosines, while thymine residues represent unmethylated cytosines in the original DNA sample. Whole genome sequencing of bisulfite converted (WGBS) DNA samples enabled genome-wide DNA methylation profiling and was employed to establish genome-wide methylation maps. However, the harsh conditions of bisulfite conversion in combination with the already fragmented population of cfDNA molecules is problematic and might result in significant sample loss [46]. A bisulfite conversion-based approach adjusted for low input DNA is the post bisulfite adaptor tagging (PBAT) technique, which is a PCR-free method that reduces bisulfite induced DNA degradation by attaching adaptors to DNA after bisulfite treatment [47,48]. An adaptation of the PBAT technique for the analysis of methylation status at single cell level by single cell bisulfite sequencing (scBS-seq) [49,50] together with the PBAT method might be employable for cfDNA analyses.

Other genome-wide approaches combining the use of enzymatic and chemical modifications are enzymatic methyl-sequencing (EM-seq) [51,52] and ten-eleven translocation (TET)-assisted pyridine borane sequencing (TAPS) [53]. These 5 mC (and 5 hmC, see last part of chapter) mapping techniques are bisulfite-free and might be an alternative for plasma samples. Recently, a cfDNA-adapted protocol for TAPS was published [54].

An alternative genome-wide method for mapping methylation is immunoprecipitation of methylated DNA (MeDIP) with 5-methylcytidine-specific antibodies and subsequent sequencing of captured genomic regions [55]. This sensitive bisulfite-free workflow is especially applicable for low input samples. The original MeDIP protocol was optimized for even lower input [56], such as cfDNA from plasma, and enables detection of large-scale DNA methylation changes that are enriched for tumor-specific patterns. Immunoprecipitation of methylated DNA can further be achieved via proteins containing a methyl-binding domain (MBD). This technique followed by sequencing of captured DNA is called MBD- or MethylCap-seq [57,58] for which a cfDNA-optimized protocol exists [59].

A reduced genome approach for DNA methylation analysis is reduced representation bisulfite sequencing (RRBS) [60]. Here, CpG-rich regions across the genome are enriched via restriction enzyme digestion followed by bisulfite conversion and sequencing. This reduced genome approach is more cost-effective compared to whole genome sequencing (WGS). This technique was also optimized for cfDNA samples (cfRRBS) [61,62]. Microarrays are another option for reduced genome methylation profiling. Commercially available microarrays facilitate DNA methylation analysis of specific, pre-selected regions of interest. This reduced cost technique is ideal for clinical applications, but prior target knowledge is required. Microarrays have been extensively employed for methylation mapping and establishing reference methylome atlases [32].

Targeted approaches for methylation mapping include targeted bisulfite amplicon sequencing (targeted BS-seq) [63,64]. Target selection can be performed by PCR amplification or probe hybridization capture. For targeted BS-seq, DNA is bisulfite converted and amplified using specific and validated primers. This technique was also optimized for plasma samples [65].

For tissue-of-origin analysis, genomic regions are identified that exhibit hyper- or hypomethylation in one cell or tissue type compared with others. These differentially methylated regions (DMRs) are subsequently employed as markers for tissue-of-origin analysis. An early study demonstrated that differentially methylated regions can be employed to detect prostate cancer [66]. In a next step, differential methylation across distinct cell or tissue types can be utilized to trace the cell- or tissue-of-origin of a single cfDNA molecule, even of different cell types within a particular tissue [67]. The basis for methylation-based tissue-of-origin analysis are high resolution methylation maps of multiple different reference cell types or tissues [68,69,70,71]. Several projects were initiated that generated high resolution methylation or epigenetic maps that are mostly available as open access datasets, such as the Roadmap Epigenomics Project [68], the ENCODE Project [72,73], the International Human Epigenome Consortium (IHEC) [74], the Cancer Genome Atlas (TCGA; https://www.cancer.gov/tcga, accessed on 11 February 2022), and the Gene Expression Omnibus (GEO) [75]. Tissue-of-origin analysis requires the development of qualified algorithms for deconvolution of sequencing data with reference methylation profiles of different tissues to determine the cell- or tissue-of-origin of cfDNA molecules and to estimate the major tissue contributors to the cfDNA pool. WGBS of plasma samples and analysis of tumor-associated hypomethylation in combination with tumor-associated copy number aberrations (CNAs) enabled the detection of several non-metastatic cancer types with a sensitivity and specificity of 87 and 88%, respectively [76]. Further, WGBS was employed to generate cfDNA methylation profiles that were used to infer relative contributions of four different tissues using a deconvolution approach [21]. For this purpose, several high resolution patterns of multiple tissue types were employed as reference methylomes [68,69,70]. The authors identified two types of methylation markers: (i) type I marker is a genomic locus that shows a methylation level in one of the tissues that is significantly different from those in other tissues and (ii) type II marker is a genomic locus with high variability in methylation densities. This approach demonstrated that ≥70% of the cfDNA pool originated from white blood cells (i.e., neutrophils and lymphocytes) in conditions in which source tissue differed genetically from host tissue (i.e., pregnancy, transplantation, cancer) and showed that the methylation deconvolution approach is able to identify the tissue-of-origin of CNAs [21]. Moreover, tissue-of-origin analysis was successfully performed with urinary cfDNA as well [77]. The authors found a high variation of proportional contribution from different tissues (i.e., neutrophils, T-cells, B-cells, urothelium, and kidney) to the urine DNA pool [77]. 

Expanding the analysis window from one CpG site to a number of adjacent CpGs led to enhanced sensitivity and reduced background of methylation analysis [22]. The authors were able to demonstrate that tissue-specific methylation patterns in cfDNA could be used to detect tissue cell death with a high level of specificity and sensitivity in multiple human pathologies. This approach was based on targeted sequencing of specific markers, which reduced sequencing costs immensely, and was able to show origins of cfDNA in pathologies such as β-cell death in diabetes, brain cell death in multiple sclerosis and head trauma without genetically distinguishable tissue [22]. A systematic search of highly coordinated methylation by a combined method approach of WGBS, RRBS, and methylation arrays identified thousands of tightly coupled CpG sites, termed methylation haplotype blocks [30]. By defining the methylation haplotype load (MHL) as a metric that is capable of quantitatively distinguishing blocks that have the same average methylation levels but various degrees of coordinated methylation, the authors were able to directly compare different regions across multiple data sets and observed a reduction in perfectly coupled CpG pairs in cancer patients. The study showed that this methylation haplotype blocks approach can be employed for quantitative estimation of tumor load and tissue-of-origin mapping of cfDNA in patients with lung or colorectal cancer [30]. The development of probabilistic models such as CancerLocator or CancerDetector enabled researchers to infer the relative proportions and tissue-of-origin of tumor-derived cfDNA from genome-wide DNA methylation data or joint methylation states of multiple adjacent CpG sites. This allowed the detection of cancer with high sensitivity and specificity [33,78].

Comparative methylome analysis identified several genomic loci that are unmethylated specifically in cardiomyocytes or hepatocytes, respectively, and could be utilized to quantify cardiomyocyte- or hepatocyte-derived DNA in cfDNA to detect acute cardiomyocyte or hepatocyte death [17,18]. This method is based on the analysis of a limited number of genomic loci, which reduces turnaround time tremendously and might complement existing biomarkers such as troponins or liver enzymes. 

The development of a sensitive, immunoprecipitation-based protocol to analyze the methylome of small quantities of cfDNA (cfMeDIP) marked an important step in methylation pattern analysis of low-input plasma DNA samples that are already of fragmented nature [31]. Specifically, the authors optimized an existing low-input MeDIP-seq protocol using exogenous *Enterobacteria phage λ* DNA (filler DNA) for DNA inputs as low as 1 to 10 ng. The work demonstrated that cfMeDIP-seq was able to detect large-scale DNA methylation changes that were enriched for tumor-specific patterns and robustly detect and classify cancer in plasma samples from several tumor types [31].

Tissue-of-origin analysis strongly relies on reference methylomes of key tissues. Since tissues mostly represent mixtures of distinct cell types, Moss and colleagues established a reference atlas of 25 human tissues and cell types covering major organs and cells involved in common diseases [32]. They employed methylation microarrays and demonstrated that plasma methylation patterns could be used to accurately identify cell type-specific cfDNA in healthy and pathological conditions including islet transplantation, sepsis, and cancer of unknown primary [32]. Further, they quantified the major cell types contributing to cfDNA in healthy individuals with 55% originating from white blood cells, 30% from erythrocyte progenitors, 10% deriving from vascular endothelial cells and 1% was hepatocyte-derived [32]. An in silico approach using automated machine-learning conducted differential methylation analysis on available microarray methylomes to establish low number biosignatures (i.e., several differentially methylated genes) for breast cancer, osteoarthritis, and diabetes mellitus and achieved high performance [79]. In order to detect distant metastases and potentially advance early detection, cell type-specific cfDNA methylation markers were recently used to identify collateral tissue damage in cancer [80]. In this work, elevated levels of hepatocyte-derived cfDNA were detected in the plasma of patients with liver metastases vs. cancer patients without liver metastases. Moreover, patients with brain metastases were identified by increased levels of neuron-, oligodendrocyte-, and astrocyte-derived cfDNA [80].

Recently, the single-molecule real-time (SMRT) long-read sequencing technology by Pacific Biosciences was modified to accurately detect 5 mC modifications [81]. This method uses a convolutional neural network to analyze the sequence context and pulse signals associated with DNA polymerase kinetics for accurate detection of 5 mC modifications and might be employed as alternative technique for methylation pattern analysis. Most recently, a human methylome atlas based on deep WGBS and 39 cell types sorted from healthy tissue samples was completed [34]. Loci uniquely unmethylated in a specific cell type are often located at transcriptional enhancers and contain DNA binding sites for tissue-specific transcriptional regulators, whereas uniquely hypermethylated loci are rare and enriched for CpG islands, polycomb targets and CTCF binding sites [34]. The authors developed a computational machine-learning suite to represent, compress, visualize and analyze WGBS data (available at: https://github.com/nloyfer/wgbs_tools, accessed on 7 February 2022) [34]. The Circulating Cell-free Genome Atlas study aimed at establishing a blood-based multi-cancer early detection (MCED) test utilizing cfDNA targeted methylation-based sequencing in combination with machine-learning in order to detect cancer signals across multiple cancer types and predict cancer signal origin with high accuracy [35,65,82]. The clinical validation study concluded that the MCED test demonstrated high specificity and accuracy of cancer signal origin and the test detected cancer signals across many different types of cancers. These results support that this minimally invasive MCED test complements and expands existing single-cancer screening tests [35,65,82].

For methylation patterns, there are currently two bioinformatics approaches that are employed: (i) the reference-based approach [21,22,30,32,34], and (ii) the reference-independent approach [31,83,84]. Both methods rely on WGBS, RRBS, or microarray methylation data to detect DNA methylation at single-nucleotide resolution genome-wide or at thousands of sites in the case of microarrays. For the reference-based approach, differentially methylated regions or positions (DMRs or DMPs) that are unique to a specific cell type are identified (i.e., methylated in a specific cell type and unmethylated in all other cell types or vice versa). Quadratic programming (QP; also referred to as constrained projection(CP)) is a well-established methylation-based bioinformatic approach that infers proportions of cell types present in the reference DNA methylation database using a constrained projection via least-squares minimization [67]. Other reference-based algorithms are CIBERSORT, an advanced Support Vector Regression for penalized multivariate regression after estimating the regression weights, or robust partial correlation (RPC) [85,86]. For these algorithms to perform well, high quality methylation atlases [32,34] based on samples comprising of preferably one cell type are required to select appropriate markers/loci that are robust enough to differentiate between distinct cell types in a mixture of unknown cell types. There is currently a trend towards broader sequencing techniques called ‘third generation liquid biopsy’ [87,88]. Techniques such as WGBS lead to higher sensitivity by using many markers—also called “features”. Therefore, feature selection is becoming increasingly important to reduce the number of markers, which in turn facilitates the interpretation of the data. Feature selection can be guided by databases such as the NCBI Gene Expression Omnibus [75] to learn which loci are reliable [32]. Other groups, such as Loyfer et al., exploited the inherent sequential data structure of the genome by grouping features that were adjacent and co-varied by cell type using a dynamically programmed probabilistic Bayes model [34]. Their aim was to obtain homogeneous blocks of methylated or unmethylated markers to (i) perform dimensionality reduction and (ii) increase the robustness of the markers. The blocks were then selected by the number of markers per block to reduce 7.2 million blocks to 2.1 million blocks. Based on this data set, Loyfer et al. filtered out further characteristics that they needed for their research goals. They arrived at a minimum of about a thousand blocks for their atlas [34]. The reference-based approach is limited to tissue and cell types that can be purified sufficiently. Furthermore, a reference in an atlas needs to be measurable and the approach does not account for influences from other cell types on the expression profile of the investigated cell type. On the other hand, quantification of cell type contributions at the single sample level is possible and the algorithms are relatively assumption-free. Reference-free algorithms include EWASher [89], RefFreeEWAS [90], ReFACTOR [91], the removing unwanted variation (RUV) framework [92], surrogate variable analysis (SVA) [93], independent surrogate variable analysis (ISVA) [94], or are based on non-negative matrix factorization (NMF) [95,96]. The reference-independent algorithms do not require reference methylation profiles or prior knowledge of cell types, and are therefore applicable to any tissue type, and account for cell-cell interactions. However, the performance of reference-independent algorithms strongly depends on whether the model assumptions are valid and sample-specific estimates of cell type fractions are not possible.

In addition to the extensively scrutinized 5 mC modification, 5-hydroxymethylcytosine (5 hmC) is an intermediate DNA modification that influences biological processes ranging from development to pathogenesis [97,98]. The ten-eleven translocation (TET) family dioxygenases convert 5 mC into 5 hmC and this DNA modification is generally thought to reflect gene activation on permissive chromatin [99]. It is particularly located at enhancers, gene bodies and promoters and changes in 5 hmC reflect alterations in gene expression levels [100,101]. 5 hmC additionally displays a tissue-specific mass distribution [102,103] and decreased levels of 5 hmC are often observed in many solid tumors compared with corresponding healthy tissues [104]. Several groups demonstrated that 5 hmC signatures in cfDNA can be utilized to detect cancer type and stage [105,106,107]. One study demonstrated that pancreatic ductal adenocarcinoma tissue-derived hyper-hydroxymethylated genes can separate non-cancer cfDNA from PDAC cfDNA samples [105]; making hydroxymethylation patterns another promising approach for early detection and tissue-of-origin analysis. 

Taken together, deep sequencing methods such as WGBS and the development of deconvolution algorithms paved the way for methylation-based tissue-of-origin analysis of cfDNA to detect various cancer types and numerous other pathologies. Employment of targeted approaches and methodological advancements such as immunoprecipitation of methylated DNA (MeDIP) reduced the costs significantly, which is a prerequisite for clinical application. However, for methylation-based tissue-of-origin analysis to become routine application the turnaround time and workflow need to be optimized and larger validation studies are required.

### 2.2. Nucleosome Positioning-Based Tissue-of-Origin Analysis

DNA is packaged into chromatin by several histone proteins in order to fit within the nucleus. The degree of DNA compaction regulates its accessibility and gene expression. The histone protein octamer with ~147 bp of DNA wrapped around it is termed a nucleosome and represents the basic unit of DNA compaction. Another histone protein—H1—binds to ~20 bp of linker DNA adjacent of the nucleosome core and the ~167 bp DNA with the histone protein octamer and H1 is called chromatosome. Nucleosomes are spaced with varying distances to each other along the DNA strand and DNA regions tightly occupied by nucleosomes are rather inaccessible, whereas DNA regions with a wider spacing of nucleosomes are more accessible for DNA binding proteins. By hindering proteins involved in gene regulation and transcription from binding, nucleosomes regulate DNA accessibility and transcription. Additionally, for TF binding and gene expression it is essential that nucleosomes move along the DNA or are removed for chromatin opening at specific genomic regions [108,109,110]. Gene expression is unique for every cell type, differing considerably between distinct cell types and tissues, and consequently nucleosome positioning varies significantly between different cell types [20]. Upon release into a body fluid, nuclear DNA is cleaved by distinct enzymes dependent on cell type and release mechanism [25]. Open chromatin regions lacking nucleosomes are poorly protected against digestion compared with nucleosome-occupied DNA regions. Thus, nucleosome-bound regions are expected to be found more frequently in the plasma DNA pool than nucleosome-depleted regions. Transcription-prone regions with low nucleosome occupancy are underrepresented in sequencing data of cfDNA, whereas higher coverage suggests lower expression levels of this genomic region. Nucleosome positioning is consequently reflected by sequence read-density across the genome and can be employed to extract information about gene expression and thereby cell identity [111], enabling tissue-of-origin analysis via cfDNA read frequency pattern.

Several accessibility assays exist to characterize nucleosome positioning and examine nucleosome architecture and gene regulation [112]. Most techniques employ enzymatic digestion or mechanical shearing of DNA of interest and subsequent analysis of resulting fragments. DNase-seq involves enzymatic digestion by DNase I that preferentially digests open regions of chromatin, leaving behind mostly nucleosome-bound DNA regions that are subsequently sequenced [113]. Seldomly found regions in DNase-seq are called DNase I hypersensitive sites (DHSs) and reflect active regulatory regions. Micrococcal nuclease (MNase) is another endo-exonuclease that preferentially digests unprotected DNA (i.e., accessible linker DNA between nucleosomes) and mostly leaves nucleosome-occupied DNA intact. MNase assays followed by sequencing were performed to identify nucleosome occupied regions of DNA [114], showing pronounced nucleosome-depleted regions (NDRs) directly upstream of the transcription start site (TSS) at promoters of highly expressed genes [108,115,116]. Another NDR was found at transcription termination sites (TTSs) of highly expressed genes [115]. Pronounced NDRs are not detected at genes with no or low transcription activity. Many nucleosome position maps were generated via the MNase assay. Formaldehyde-assisted isolation of regulatory elements sequencing (FAIRE-seq) utilizes crosslinking of chromatin-interacting proteins to DNA by formaldehyde, chromatin shearing and subsequent phenol-chloroform extraction to separate DNA unbound by proteins (aqueous layer) from protein-DNA crosslinks (organic layer) [117]. An assay for transposase accessibility sequencing (ATAC-seq) employs hyperactive Tn5 transposase to insert sequencing adapters at accessible regions of the genome that will subsequently be sequenced [118,119]. Thereby, resulting sequencing reads represent regions of increased accessibility and can be used to map nucleosome positions.

To determine the nucleosome positions by analysis of plasma DNA, no experimental enzymatic digestion is necessary due to the fragmented nature of cfDNA. Nucleases that are present intracellularly and in the blood degrade preferentially DNA regions that are not bound by nucleosomes or other proteins. Because nucleosome-bound DNA is protected better against degradation, (deep) sequencing of the total plasma DNA pool primarily yields reads of nucleosome-bound regions. By aligning the resulting sequencing reads of a plasma sample to the reference genome, nucleosome positions can be inferred from and information on the cell identity can be extracted.

One study built a genome-wide map of in vivo nucleosome occupancy of cfDNA based on deep sequencing of total cfDNA and demonstrated that the resulting nucleosome spacing pattern could be utilized to perform tissue-of-origin analysis [20]. Specifically, they defined a metric—the windowed protection score (WPS)—to determine the nucleosome occupancy at a given genomic coordinate. For this purpose, they looked at 120 bp windows and quantified how many fragments (120–180 bp) ended in that window and how many fragments completely spanned it. By subtracting the number of fragments ending in that window from the number of fragments completely covering that window, they yielded the L-WPS of any genomic coordinate. A higher WPS is correlated with many fragments spanning that specific window and few fragments ending in that window, indicating that this DNA region is occupied by a nucleosome or other protein. Vice versa, a lower L-WPS corresponds to a low number of fragments spanning the window and a higher number of fragments ending in that window, suggesting that this region is not protected by a nucleosome. The metric yielded a graph that spanned the entire genome and that was analyzed for local maxima that represented regions occupied by nucleosomes, resulting in approx. 10 million peaks. To verify this hypothesis that WPS peaks represented regions occupied by nucleosomes, the authors used ChIP-seq data from the ENCODE database [120] to adjust the filtering to only receive high confidence peaks. The fragment length was the main aspect for inferring nucleosome positions in Snyder et al. and resulted in a specific pattern that was distinct for different cell types and genomic regions [20]. Additionally, single-stranded libraries revealed shorter DNA fragments that directly footprinted TF occupancy (for details see Section 2.3). Those epigenetic footprints matched hematopoietic lineage in healthy individuals, while additional contributions were observed in cancer patients, often aligning with the cancer type [20]. 

Another group reported that gene expression could be predicted with plasma DNA coverage in promoter regions [37]. The authors developed a nucleosome promoter analysis utilizing machine-learning and WGS datasets to determine the expression status of a gene. For this purpose, they used coverage at the TSS to infer gene expression from nucleosome occupancy [37]. They defined two ranges: (i) 2 kbp centered around the TSS and (ii) −150 bp to 50 bp with respect to the TSS. Applying kernel density estimation to the normalized read coverage in these two ranges led to the distinction between two clusters. These two clusters could be classified with an accuracy of 0.91 using a support vector machine [121] of the most and least expressed 100 genes [37]. This approach identified two discrete regions at TSSs where nucleosome occupancy resulted in different read depth coverage patterns for expressed and silent genes, allowing classification of expressed cancer driver genes in regions with somatic copy number gains in patients with metastatic cancer [37]. 

A compilation of published in vivo nucleosome positioning datasets in a database called NucPosDB provides a comprehensive overview on available datasets and includes datasets of sequenced cfDNA that reflect nucleosome positioning in situ in the cells of origins [122]. Moreover, a list of computational tools for the analysis of nucleosome positioning or cfDNA experiments can be found at NucPosDB and the database contains theoretical algorithms for the prediction of nucleosome positioning preferences from DNA sequence [122]. Different approaches employing nucleosome footprinting for tissue-of-origin analysis of cfDNA fragments determined varying contributions of white blood cells and other organs to the plasma DNA pool [20,21,123,124,125]. 

The combination of EM-seq to study methylation patterns in combination with analyzing nucleosome occupancy in a single assay (cell-free DNA-based nucleosome occupancy and methylation profiling: cfNOMe) might advance tissue-of-origin analysis in plasma samples [126].

A better understanding of the underlying release mechanisms and cfDNA biology is pivotal to unravel the wealth of valuable information encoded in nucleosome positions.

### 2.3. Tissue-of-Origin Analysis by Inferring Transcription Factor Binding Sites

TFs are essential factors in regulating gene expression by promoting or blocking the recruitment of RNA polymerase to specific genes. Thereby they fine-tune the expression of their target genes and are key players in development and differentiation [127]. By failing to suppress cancer driver genes or silencing tumor suppressor genes, dysfunctional TFs can lead to carcinogenesis. As transcription differs tremendously between distinct cell types, distinct TFs are involved in gene expression depending on the cell type. Identifying occupied transcription factor binding sites (TFBS) within plasma DNA might be a way for tissue-of-origin analysis, allowing a more dynamic snapshot on rapidly changing gene expression in response to cancer or other pathologies.

To determine accessible DNA regions, the afore-mentioned techniques for studying accessible chromatin/nucleosome occupancy can be employed (see Section 2.2). In the case of cfDNA, enzymatic digestion is not needed due to the fragmented state of cfDNA. Sequencing cfDNA and aligning the obtained reads to the human genome results in sequence read density across the genome. Regions with low read density are suggestive of DNA lacking TFs or nucleosomes, while regions with high read density indicate TFs or nucleosomes bound to that region. However, due to the smaller size of TFs, TFBS occupancy is more challenging to determine experimentally. Consequently, few studies have to date attempted to infer TFBS occupancy from cfDNA fragmentation patterns.

One study employed single-stranded library preparation of cfDNA samples to recover short fragments and used deep sequencing to investigate if it is possible to utilize TFBS occupancy for tissue-of-origin analysis. The authors observed that short cfDNA fragments (35–80 bp) directly footprint the in vivo occupancy of TFBSs of several TFs [20]. To demonstrate this, they determined the occupancy of a TFBS by defining the short fragment windowed protection score (S-WPS) at a given genomic coordinate. The S-WPS is the number of short DNA fragments (35–80 bp) completely spanning a 16 bp window centered at a given genomic coordinate subtracted by the number of fragments with an endpoint within that window. By focusing on short cfDNA fragments, they were able to infer additional contributing tissues to the plasma DNA pool in non-healthy states [20]. The bioinformatic analysis occurred identically to the L-WPS analysis. Ulz et al. employed publicly available ATAC-seq data and WGS data from plasma cfDNA to infer accessibility of hundreds of TFBSs from cfDNA fragmentation patterns [29]. To determine TFBS occupancy, they developed a metric called accessibility score that measured the strength of TF phasing at the TFBS, reflecting the strength of the TF binding. [29]. This score was based on the read coverage quantifying the amplitudes close to TFBSs and then applying signal filtering with the Savitzky-Golay algorithm [128] to smoothen the graph, making the amplitudes easier to measure and detect. The rank differences (i.e., overall z-scores) in this high frequency signal between tumor and healthy samples with defined thresholds for TFBS accessibility differences were used as the accessibility score. A logistic regression was applied based on 504 markers that were pre-selected. A cross-validation approach was used to train and test the algorithm. By quantifying the amplitudes at TFBSs, the authors were able to identify nucleosome-depleted regions (NDRs) and thereby distinguish between control and cancer samples and even different subtypes of cancers. Applying this approach, the authors were able to profile numerous individual TFs and objectively compare TF binding events in plasma samples, allowing subclassification of tumor entities and TFBS plasticity during disease progression [29].

### 2.4. Tissue-of-Origin Analysis Utilizing Histone Modifications

Histone proteins involved in DNA compaction can be post-translationally modified in multiple ways, including methylation, acetylation, phosphorylation, adenosine diphosphate (ADP) ribosylation, ubiquitylation, sumoylation, formylation, and hydroxylation of specific histone amino acids [129,130]. These various modifications change the interaction between DNA and nuclear proteins (i.e., by compelling or attracting DNA from or to histone proteins, respectively) or modify the binding affinity of chromatin remodelers or the transcription machinery to the nucleosome. Thereby, they regulate chromatin accessibility and gene transcription and can be grouped into four major functions: activating, repressing, heterogeneous, and bivalent modifications. Methylation and acetylation are among the most well-studied histone modifications. The addition of an acetyl group influences chromatin compaction by neutralizing the basic charge at unmodified lysine residues and weakens the electrostatic interaction between the negatively charged DNA and histones [131]. Histone acetylation has further been implicated in regulation of the intracellular pH [132]. Histone acetylation is largely associated with active transcription, particularly when located at enhancers, promoters, and the gene body [133]. Histone methylation, on the other side, retains the basic character of the modified histone and represents a subtler modification [134]. Trimethylation of histone H3 at lysine residues 9 and 27 (H3K9me3 and H3K27me3) for instance has a repressive effect on transcription and marks regions of closed chromatin [129], while (tri-) methylation of histone H3 at lysine residues 4 and 36 (H3K4me1/2/3, and H3K36me1/2/3) and acetylation of histone H4 at lysine 16 (H4K16ac) are considered activating marks and hint at open chromatin regions where active transcription occurs [129]. Histone modification patterns mirror recent and transient changes in chromatin regulation and RNA polymerase activity, often occurring at the onset of pathologies when cells attempt to adapt to altered conditions by changing their transcriptional profiles. Accessible and active promoters, enhancers and gene bodies of actively transcribed genes are characterized by the presence of specific combinations of histone modifications [135,136,137,138,139,140]. Hence, histone modifications are indicative of altered transcriptional programs in response to pathologies. Specifically, changes in histone modification patterns have been extensively linked to cancer [134]. Hyperacetylation, particularly at proto-oncogenes, may activate gene expression. Conversely, hypoacetylation of tumor suppressor genes (often localized at promoters) might silence gene expression. Post-translational methylation of histones represents a functionally very complex regulation mechanism and both elevations and reductions in histone methylation have been associated with, e.g., cancer.

The most commonly used technique for profiling protein-DNA interaction is chromatin immunoprecipitation (ChIP) [141,142]. First, crosslinking is performed to covalently bind the lysines of interacting proteins with local DNA. Second, cell lysis and subsequent shearing by sonication are carried out, followed by incubation with an antibody targeting the protein of interest or the specific histone modification of interest and finally the interacting DNA region is pulled down, usually via beads attached to the secondary antibody, crosslinks are reversed, protein is digested and sequencing is performed. One drawback of this method is, however, the high input needed to produce a high signal-to-noise ratio. Adaptations to the original protocol reduced the input need (µChIP) [143]. Employing MNase digestion instead of sonication (native ChIP) further decreased the DNA input required [144]. However, applying ChIP to very low-input plasma samples remains technically challenging. Recently, the chromatin immunoprecipitation protocol was modified specifically for low-input cfDNA from plasma samples (cfChIP) [28].

cfChIP followed by sequencing (cfChIP-seq) demonstrated that plasma nucleosomes maintain the epigenetic information of the cell they originated from and that cfChIP-seq recapitulates the original genomic distribution of histone modifications related to active transcription. Antibodies specific for different histone modifications, which were immobilized on paramagnetic beads, were directly incubated with plasma samples. Subsequently, on-bead adapter ligation was performed before DNA isolation. Four antibodies specific for active transcription (accessible/active promoters: H3K4me3 or H3K4me2; accessible enhancers: H3K4me2; gene bodies of actively transcribed genes: H3K36me3) were employed in this study that included plasma samples from healthy individuals, patients with different liver diseases, and patients with metastatic colorectal carcinoma. The authors showed that cfChIP-seq allowed genome-wide unbiased analysis and was capable of determining the tissue-of-origin and detecting differences in patient- and disease-specific transcriptional programs (including cancer-specific signatures) by generating biologically relevant reduced representation of the genome [28]. Individual tailored bioinformatics procedures were applied for each of the four antibodies used in this study. The analysis showed that bone marrow megakaryocytes were identified as major contributors to the plasma DNA pool in healthy individuals and pathology-related changes in hepatocytes chromatin were found in patients with liver diseases. 

Colorectal cancer was detected with high sensitivity using cfChIP-seq and could identify subgroups of patients with distinct cancer-related transcriptional programs [28]. Another study employing blood plasma cfChIP revealed that H3K36me3 cfChIP followed by droplet digital PCR can be used to identify tumor-specific transcriptional activity of the mutated EGFR-L858R allele in non-small cell lung cancer [145]. This focus on tumor-specific transcriptional activity of genes harboring somatic mutations will help to gain more insights on the relevance of mutations in, e.g., therapy resistance mechanisms. 

So far, however, only activating histone modifications for tissue-of-origin analysis have been studied. Prospectively, it will be of great interest to investigate repressive and heterogeneous modifications as well. It might also be helpful to look into combinations of distinct histone modifications, e.g., immunoprecipitate a specific histone acetylation prior to a specific histone methylation. Hereby, it is essential to employ validated antibodies (e.g., by IHEC) of sufficient quality and to streamline the different existing protocols to achieve standardization and comparability of studies performed by different groups. Importantly, cfChIP-seq provides information on transient changes in gene expression altered upon various pathologies and during disease progression.

### 2.5. Tissue-of-Origin Analysis Based on Fragmentomics

Nucleosome occupancy and DNA compaction degree are not the only determinants of cfDNA fragment size, but the release pathway and nucleases present in blood also play major roles [146,147]. Depending on the cell type, release mechanism (e.g., apoptosis or necrosis), and DNA condensation mode, nuclear DNA is cleaved by distinct nucleases, resulting in varying cfDNA fragments that are characteristic for each cell type [11]. DNA fragments bound to proteins (typically histones or TFs among other proteins such as albumin, HDL, etc.) are preferentially found in bodily fluids, while naked DNA is mostly digested [148]. Commonly, the majority of cfDNA molecules exhibit the size of DNA wrapped around one nucleosome plus DNA linker (i.e., ~167 bp) [20,23,27]. A series of additional peaks with ~10 bp periodicity below ~143 bp putatively correspond to the helical pitch and binding sites of DNA to the nucleosome core [20]. Size distribution of cfDNA fragments is altered by pathologies, e.g., cfDNA from cancer patients was found to be slightly shorter than cfDNA from healthy individuals (147 vs. 167 bp) [23,27], which is possibly explained by a different binding strength of the external histone H1. Generally, the lengths of cancer-derived cfDNA fragments have a tendency to be more variable than non-cancer DNA [23]. In addition to mono-, di-, and trinucleosome-sized cfDNA fragments predominantly found in plasma samples, considerably longer cfDNA fragments were detected, particularly a ~3 kbp cfDNA population (ranging between 1–6 kbp with an average size of 3 kbp) initially thought to originate from randomly lysed cells that might represent genomic DNA contamination [149,150,151,152,153]. Currently, many groups hold the view that this ~3 kbp cfDNA population is actively extruded by live cells [154,155,156,157,158,159] and it might represent an artifact of incomplete size separation with an underlying DNA laddering pattern of at least seven nucleosomes or more [160]. Depending on the nuclease(s) involved in the release pathway, the cfDNA fragments exhibit distinct fragment end motifs (i.e., four-base motif at both ends of a cfDNA fragment) [39] with “CCCA” being the most common 4-mer end motif in plasma DNA fragments from healthy individuals [161]. DNASE1L3 was suggested to generate predominantly end motifs starting with “CC” [24,162], DNA fragmentation factor subunit β (DFFB) was found to be responsible for 5′ “A” end motifs [25], and DNASE1 activity was associated with generating “T” end motifs [25]. Nuclease activity is thought to be altered upon pathogenesis and might be utilized for tissue-of-origin analysis. 

In particular, cfDNA size distribution, preferred ends (i.e., genomic coordinates that are found at cfDNA ends more often than others), end motifs of cfDNA, end orientation of cfDNA, and topology of ends (i.e., double-stranded vs. single-stranded) are informative about tissue-of-origin and numerous studies demonstrated the feasibility of fragmentomics-based liquid biopsies to detect cancer and other pathologies [23,26,27,38]. These fragmentomics features can be studied by sequencing plasma samples and analyzing the fragmentomics feature of interest. However, to determine fragmentomics features of cfDNA molecules accurately, it is essential to preserve the charactistics of plasma DNA molecules by a preanalytical routine that does not alter fragmentation and by a gentle and suitable library preparation procedure (e.g., optimized single-stranded library preparation or a library preparation method without end repair). Thus, the cfDNA fragmentomics features can be employed to distinguish between healthy and diseased cells. Pathologies and injuries have in common that they induce increased cell death in the affected tissue, leading to elevated levels of cfDNA molecules to the plasma DNA pool from the pathogenic tissue. Tissue-of-origin analysis based on fragmentation patterns have been shown to reveal elevated contributions to the plasma DNA pool and enable sensitive detection of pathologies lacking genetic differences, such as myocardial infarction, stroke and autoimmune disorders.

An early study on plasma cfDNA fragmentation patterns demonstrated that fragmentation occurs non-randomly and cfDNA retained characteristics previously found in genome-wide analysis of chromatin structure and are concordant with corresponding cell-line derived patterns [163]. The same authors further investigated fragmentation patterns of cfDNA from cerebrospinal fluid (CFS) from glioma patients and found a distinct fragmentation pattern of cfDNA in CFS [164]. Another research group developed an approach to detect the total pool of cancer-associated somatic mutations in plasma and studied the cfDNA end characteristics in patients with hepatocellular carcinoma and chronic hepatitis [26]. The authors found tumor-associated cfDNA preferred end-coordinates at certain genomic coordinates that point to cfDNA derived from transplanted liver, hepatocellular carcinoma, or the placenta. The number of cfDNA molecules with end signatures for tumor or liver correlated with the amounts of tumor- or liver-derived cfDNA in plasma [26]. The comprehensive analysis of cfDNA fragmentation patterns in plasma samples from patients with different cancer types and healthy individuals identified differences in the size distribution of tumor-derived and non-cancer DNA fragments [23], namely mutant ctDNA was more fragmented than non-mutant cfDNA. This observation led the authors to develop an approach that selectively analyzes short cfDNA fragments in the size of 90 to 150 bp with a machine-learning algorithm, achieving greater sensitivity to detect tumor DNA from multiple cancer types in plasma [23]. Extending the fragmentation pattern analysis to additionally determining cfDNA fragment orientation (i.e., upstream or downstream fragment end profile) can identify short linker DNA and tissue-specific open chromatin regions, allowing determination of relative contributions of various tissues to the plasma DNA pool [38]. 

Evaluating fragmentation patterns across the genome at megabase level (the authors looked at ~500 windows of 5 Mbp size each) to observe large-scale fragmentation patterns was the aim of a different approach termed DNA evaluation of fragments for early interception (DELFI) [27]. The authors used shallow coverage WGS data and developed a machine-learning model that incorporated genome-wide fragmentation features and examined fragmentation patterns from healthy and cancer samples. For this classifying approach, the authors utilized the gbm (stochastic gradient boosting) machine-learning model and performed a 10-fold cross-validation. They determined the ratio of short (100–150 bp) and long fragments (151–220 bp) for these 500 Mbp windows and normalized for GC content via a LOESS smoother [165]. They simulated the limit of detection based on the fraction of ctDNA in the plasma DNA pool and attempted to improve performance by adding copy number changes, chromosomal arm changes or mitochondrial DNA, but the genomic profile as defined originally performed better or equal. They observed that profiles of healthy individuals reflected nucleosomal patterns of white blood cells and patients with cancer exhibited altered fragmentation patterns. Applying the DELFI algorithm to samples from different cancer types could identify the tissue-of-origin of cancers in three-quarters of examined samples. Combining DELFI with mutation-based cfDNA analysis detected 91% of patients with cancer [27]. 

Studying the diversity of plasma DNA end motifs (i.e., the first 4-nucleotide sequence on each 5′ end of the Watson and Crick strand) and determining the motif diversity score (MDS) identified differences in the end motifs between patients with different cancers (including hepatocellular carcinoma) and healthy subjects [39]. Additionally, it was observed that plasma DNA molecules from liver, hepatocellular carcinoma, placenta, hematopoietic cells bore characteristic plasma DNA end motifs that could be utilized for tissue-of-origin analysis [39]. For instance, the end motif “CCCA” was less frequently found in samples from patients with hepatocellular carcinoma than from healthy individuals. The authors found a downregulation of DNASE1L3 in hepatocellular carcinoma cells and hypothesized that decreased DNASE1L3 activity was responsible for decrease of “CCCA”-bearing cfDNA fragments in plasma of patients with hepatocellular carcinoma. 

Moreover, cfDNA fragment end sequence patterns were investigated and utilized for tissue-of-origin analysis [166]. This work focused on the diversity of bases at the ends of cfDNA fragments (i.e., cfDNA termini) and defined a quantitative metric referred to as the fragment end integrated analysis (FrEIA) score to objectively compare fragment ends. For comparison, it was focused on the first mapped 5’ trinucleotide, the first mapped 5’ mononucleotide and the last mapped 3’ nucleotide [166]. These nucleotides were categorized into the first 5’ trinucleotide, the first 5’ nucleotide and the first and last 5’ nucleotide pair. Fractions for the fragment categories were calculated. Using these fractions, the FrES entropy (i.e., information content) was calculated by applying the normalized Shannon entropy and the Gini index. The FrEIA score developed in that study was defined as the ratio of increased trinucleotides and decreased trinucleotides in cancer multiplied by the previously mentioned normalized entropy. The machine-learning model for cancer classification was cross-validated 10-fold and included hyper-tuning and applying several different models resulted in the selection of a support vector machine (SVM) [121] as the best model. In addition, unsupervised machine-learning was employed using t-sne for clustering via k-means. The limit of detection was determined by simulating down-sampled populations [166]. By compiling a genome-wide catalogue of cfDNA fragment end sequence patterns of a large cohort of cancer patients, the authors demonstrated that fragment-end sequence and diversity were altered in 18 distinct cancer types. Furthermore, they were able to classify cancer samples from controls at low tumor content [166].

Most recently, a novel approach for analyzing expression based on cfDNA fragmentomics was developed that measured fragment length diversity to infer RNA expression levels at individual genes [40]. The working hypothesis was that cfDNA fragments from active promoters would display more random cleavage than fragments from inactive promoters due to the difference in nucleosome occupancy. To test this, targeted sequencing of promoters of genes of interest was performed and fragment length diversity was determined by promoter fragmentation entropy (PFE). The developed prediction model was based on two features: (i) promoter fragmentation entropy (PFE) that measured the diversity in fragment length distribution at a TSS compared to control genes and (ii) NDR defined as normalized counts per million of 2 kbp around the TSS. For classification of cancer vs. control and subtype classification, the authors employed logistic regression with regularization (elastic net; between 50–150 features per model). They performed cross-validation and preanalytic checks and in silico simulation for determining the limit of detection. Employing this analysis method to plasma samples could classify subtypes of lung carcinoma and diffuse large B cell lymphoma. Further, it was possible to correlate gene expression profiles with clinical response in serial blood samples from patients with PD-(L)1 immune checkpoint inhibitors treatment [40].

It has become increasingly clear during the last years that another cfDNA population exists in plasma that might exhibit diagnostic potential: ultrashort single-stranded cfDNA fragments up to 100 bp [20,167,168,169,170]. These fragments are often omitted by conventional library preparation due to the adapter ligation process that needs double strands and/or applied size selection procedures. An early study employing single-stranded library preparation and subsequent sequencing demonstrated that short cfDNA molecules directly footprint TFBS occupancy [20]. Recent efforts optimizing different DNA extraction and library preparation procedures independently demonstrated the presence of this novel population of cfDNA molecules that might inform on gene regulatory regions and DNA secondary structures [168,169,170].

Overall, cfDNA fragments hold an abundance of information that can be utilized for tissue-of-origin analysis (reviewed in detail in [171]) with broad potential clinical applications after further validation.

### 2.6. Bioinformatic Analysis of Epigenetic Features of cfDNA

The basis for a rigorous and comprehensive bioinformatic tissue-of-origin analysis is the generation of high-quality data from samples collected with well-defined preanalytical and analytical procedures. Sequencing data is quality-checked, filtered and then further processed in different ways prior to downstream analysis for deconvolution of contributing cell or tissue types. We will briefly discuss the general underlying bioinformatic procedures that have to-date been utilized for tissue-of-origin analysis based on the different epigenetic features of cfDNA molecules.

In most cases, the determination of the tissue-of-origin is done either by finding the right mixture of cfDNA from the corresponding tissues by deconvolution or by classifying whether the cfDNA contains the signature of a tissue or a disease. Deconvolution is performed using either label-free methods or reference-based methods. The reference-based methods rely on tissue references and use them to find the correct tissue admixture. The most common machine-learning methods used for cfDNA classification are random forest, logistic regression and support vector machines.

With the advancement of biotechnological methods, it is now possible to perform deconvolution and classification using many features instead of focusing on a few selected mutations. This is a paradigm shift and means that the way data from liquid biopsies is processed has to be adapted. The result is an enormous amount of data points per sample that need to be processed in a way to find the right information. This part of the process is called featurization or feature engineering. It is often based on the epigenetic feature measured using, for example, a window approach to combine adjacent CpG sites into cfDNA methylation data. The main goal is generally to extract the information appropriate to the research question and thus reduce the number of data points per sample. This process is called feature selection or dimensionality reduction. Once the data have been selected, the method of deconvolution or machine-learning is applied.

Understanding biases, such as preanalytical factors and the protocol used, becomes increasingly important as most methods use references as ground truth for comparison. When ground truths differ, interpretation of the data becomes much more difficult and can lead to incorrect assumptions.

An interesting factor for cfDNA is the available epigenetic features that can be used and differ from the use of cfDNA mutations previously. In most cases, a single epigenetic feature type, such as e.g., methylation data, is analyzed, with the exception of a few papers such as Siejka-Zielinska et al., 2021, which used methylation and fragmentation data [54]. Other research groups have used a single epigenetic trait but used a different trait to verify the gathered information, such as Tang et al., 2017, using methylation data and then verifying against miRNA expression data [172].

Overall, a wide variety of tissue-of-origin analyses based on distinct epigenetic features have been developed and their great potential for screening tests have been demonstrated. However, extensive bioinformatic expertise is needed for the development and application of the different sophisticated machine-learning algorithms. Numerous comprehensive databases that are publicly available await to be further analyzed (e.g., Roadmap Epigenomics Project [68], the ENCODE Project [72,73], the International Human Epigenome Consortium (IHEC) [74], the Cancer Genome Atlas (TCGA; https://www.cancer.gov/tcga, accessed on 11 February 2022), the Gene Expression Omnibus (GEO) [75], the Gene Transcription Regulation Database (GTRD; http://gtrd.biouml.org, accessed on 8 June 2022), the Genome-Tissue Expression (GTEx), the BLUEPRINT Epigenome (https://www.blueprint-epigenome.eu/, accessed on 8 June 2022)). Combining different approaches based on epigenetic features of cfDNA might advance minimally invasive liquid biopsies.

## 3. Conclusions and Future Perspectives

Liquid biopsies based on the analysis of epigenetic cfDNA features significantly advanced the scope of this minimally invasive approach towards a more sensitive early detection of multiple types of cancer and detection of various pathologies beyond cancer (e.g., autoimmune disorders, organ pathologies, systemic inflammation). Analysis of methylation patterns, nucleosome footprints, histone modifications, and the emerging field of fragmentomics offers a yet unprecedented potential for clinical implementation of liquid biopsies, for example as screening methods for pathologies lacking reliable tests or sensitive and serial monitoring of therapy success. However, larger validation studies, additional insights on the release mechanism as well as cfDNA biology, guidelines for standardized preanalytical procedures, and harmonized bioinformatic pipelines are urgently required prior to broad applicability of epigenetic-based liquid biopsy approaches. 

Tissue-of-origin approaches utilizing epigenetic characteristics of circulating nucleic acids are only at the early stages of development. It is not yet possible to evaluate the full potential of minimallyinvasive liquid biopsy tests for early detection and serial monitoring. As research on liquid biopsies continues to evolve, it will become clear which approach(es) or combination of approaches are best suited for detection of distinct pathologies. In addition, technical optimization of tissue-of-origin techniques will be performed as research continues. In particular, possible influences of preanalytical and analytical biases on tissue-of-origin analysis need to be further elucidated and minimized. Equally importantly, a comparative validation of the appropriate bioinformatic approach needs to be achieved. Subsequently, focus should be placed on advancing the most promising methods in order to maximize the benefit to patients. The prerequisite for clinical application is the ability of an approach to reliably stratify between healthy individuals and the patient cohort. For this purpose, all of the proof-of-concept studies presented in this review need to be further validated with larger cohorts in appropriately powered clinical trials and the workflows need to be standardized to evaluate their performance. Clinical applicability and utility for different settings (i.e., for disease identification and organ/cell type localization) need to be rigorously tested. Despite of the large amount of work that still needs to be performed, tissue-of-origin approaches based on epigenetic characteristics of plasma DNA molecules hold great promise for precision and targeted medicine.

Taken together, shifting research interest to epigenetic characteristics of circulating nucleic acids considerably boosted the performance of liquid biopsy particularly for detection of pathologies besides cancer and lays the ground for early and precision diagnostics of a broad range of diseases.

## Figures and Tables

**Figure 1 diagnostics-12-01834-f001:**
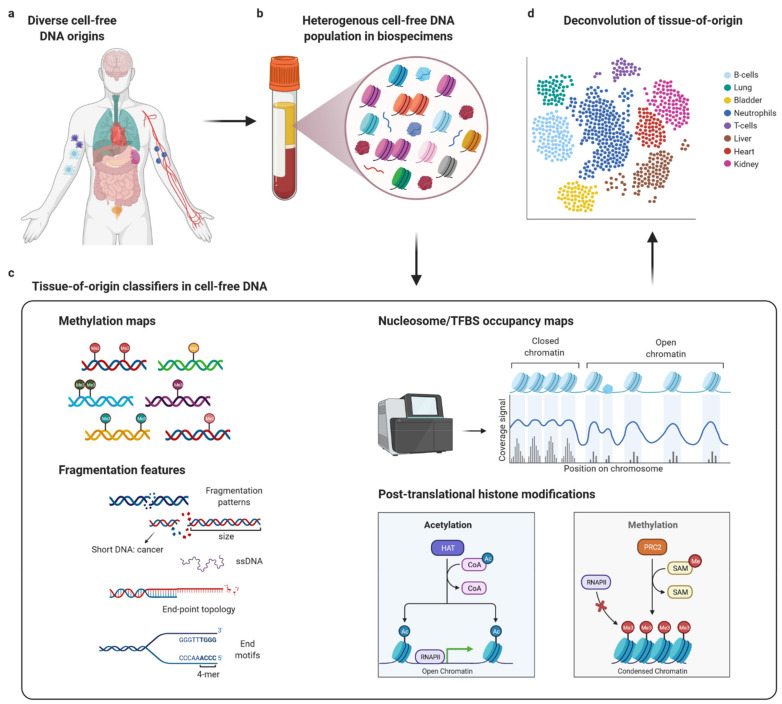
Tissue-of-origin analysis of cell-free DNA. (**a**) Different organs and various cell types release cell-free DNA (cfDNA) into blood plasma. (**b**) This clinical biospecimen represents a highly heterogeneous mixture of cfDNA molecules, often complicating the analytical differentiation between different cfDNA subtypes. (**c**) Multiple different epigenetic characteristics can be employed for tissue-of-origin analysis such as unique methylation patterns, fragmentation profiles and fragment end-points, transcription-factor binding sites occupancy, nucleosome positioning, as well as post-translational histone modifications. (**d**) Analysis of these features poses an analytical challenge, but various approaches developed recently enable determination of tissue-of-origin of individual cfDNA molecules, facilitating localization of tumors or tissue damage in specific regions such as the heart or liver.

**Table 1 diagnostics-12-01834-t001:** Overview of key studies that performed tissue-of-origin analysis on cfDNA using various approaches based on methylation patterns, nucleosome positioning patterns, TFBS occupancy, histone modifications, and fragmentomics. The data literature search was performed with the PubMed NCBI database. All deconvolution methods listed in this table are reference-based and employ a classification, unless stated otherwise.

Epigenetic Feature	Method	Approach	Disease	Deconvolution Method	References
**Methylation**	CpG islands analysis	WGBS	HCC, NIPT, Transplant	QP	[21]
**Methylation**	Analysis of adjacent CpG sites	Bisulfite amplicon-seq	PDAC, CRC, Diabetes, Transplant, MS, TBI, IBD	Read-specific binary classification	[17,22]
**Methylation**	Methylation haplotype block analysis	scRRBS, WGBS	CRC, LCP	QP, Random forest, feature extension “haplotype blocks”	[30]
**Methylation**	Analysis of differentially methylated regions (DMRs) + ctDNA abundance	CfMeDIP-seq	PDAC, AML, lung and breast cancer, CRC, RCC, bladder cancer	Limma, binomial GLM (GLMnet)	[31]
**Methylation**	Cell-type methylation atlas	Microarray	Sepsis, islet transplantation, CRC, lung, breast, prostate cancer, CUP	NNLS	[32]
**Methylation**	CancerDetector	Microarray, WGBS	Liver cancer	Maximizing log-likelihood model (grid search)	[33]
**Methylation**	Cell-type methylation atlas Analysis of blocks of homogenously methylated CpG sites	Deep WGBS	COVID-19	dynamically programmed probabilistic Bayes model, NNLS, wgbstools	[34]
**Methylation**	MCED test validation	Bisulfite amplicon-seq	12 cancer types	Ensemble logistic regression, Perceptron	[35]
**Nucleosome occupancy/ TFBS occupancy**	Windowed protection score (L-WPS/S-WPS) analysis of long/short fragments	Deep WGS	Small-cell lung cancer, squamous cell lung cancer, colorectal adenocarcinoma, HCC, ductal carcinoma in situ breast cancer	Windowed approach, Fast Fourier transformation, no classification	[20]
**Nucleosome occupancy**	Nucleosome-depleted region (NDR) analysis	WGS	Breast cancer	ABSOLUTE [36], estimation of tumor purity and ploidy	[37]
**TFBS occupancy**	Accessibility score analysis	sWGS	Prostate adenocarcinoma, breast cancer, colon adenocarcinoma	Logistic regression	[29]
**Histone modifications**	Analysis of activating histone modifications	cfChIP-seq	Colorectal carcinoma, diverse liver diseases, AMI	Robust linear regression (rlm R-package)	[28]
**Fragment size**	Fragment size distribution analysis	sWGS + in vitro size selection	High-grade serous ovarian cancer	Logistic regression, random forest	[23]
**Orientation of fragments**	Orientation-aware plasma DNA fragmentation analysis (OCF)	WGS	Pregnancy, transplant, HCC, CRC, lung cancer	Nucleosome-depletion signal used to calculate OCF value, no classification	[38]
**Fragment size**	DNA evaluation of fragments for early interception (DELFI)	sWGS + genome-wide fragmentation pattern analysis	Breast, colorectal, lung, ovarian, pancreatic, gastric, and bile duct cancer	Multi-classifying approach, stochastic gradient boosting model	[27]
**Fragment end motif**	Motif diversity score (MDS) analysis using an adopted normalized Shannon entropy	WGS	HCC, pregnancy, liver transplantation, CRC, lung cancer; head and neck squamous cell, and nasopharyngeal carcinoma	SVM, logistic regression	[39]
**Fragment size**	Promoter fragmentation entropy (PFE) analysis using a modified Shannon index	Epigenetic expression inference from cfDNA-seq (EPIC-seq)	Non-small-cell lung cancer, diffuse large B cell lymphoma	Dirichlet-multinomial model, logistic regression	[40]

Abbreviations: TFBS: transcription factor binding site; HCC: hepatocellular carcinoma; NIPT: non-invasive prenatal testing; PDAC: pancreatic ductal adenocarcinoma; CRC: colorectal cancer; MS: multiple sclerosis; TBI: traumatic brain injury; IBD: inflammatory bowel disease; LCP: lung cancer primary; AML: acute myeloid leukemia; RCC: renal cell carcinoma; AMI: acute myocardial infarction; MCEP: multi-cancer early prediction; WGBS: whole genome bisulfite sequencing; scRRBS: single-cell reduced representation bisulfte sequencing; cfMeDIP-seq: cell-free methylated DNA immunoprecipitation and sequencing; cfChIP-seq: cell-free chromatin immunoprecipitation and sequencing; GLM: generalized linear model; NNLS: non-negative least squares; QP: quadratic programming; SVM: support vector machine.

## Data Availability

Not applicable.

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
