# Peer review of "Tracing the Origin of Cell-Free DNA Molecules through Tissue-Specific Epigenetic Signatures"

_diagnostics, 2022, doi:10.3390/diagnostics12081834_

Round 1

Reviewer 1 Report

The manuscript summarized the currently available toolbox for liquid biopsies and the early detection of diseases based on the tissue-of-origin analysis.

the review is linear and well-written. However, some points need to be addressed to further improve the manuscript:

1) a table summarizing data literature should be added to have a clear picture of available data

2) please add a section with methods used to search data literature.

Author Response

Reviewer 1

Comments and Suggestions for Authors

The manuscript summarized the currently available toolbox for liquid biopsies and the early detection of diseases based on the tissue-of-origin analysis.

the review is linear and well-written. However, some points need to be addressed to further improve the manuscript:

Response: We thank the reviewer for the thorough and constructive review of our manuscript. We have created a table summarizing the key publications of tissue-of-origin analyses for a better overview and hope to meet the reviewer’s expectation.

1) a table summarizing data literature should be added to have a clear picture of available data

Response: We thank our reviewer very much for this helpful suggestion. We have created a table summarizing the key publications dealing with tissue deconvolution and tissue-of-origin analysis to give the reader a better overview on the available data.

2) please add a section with methods used to search data literature.

Response: We have now included a description on how we searched data literature in the table legend.

Reviewer 2 Report

Dear Author

Your manuscript is an interesting one but the data can be presented in a table. A table that summarized all studies on cfDNA epigenetics in several cancers.

Sincerely

Fatemeh

Author Response

Reviewer 2

Comments and Suggestions for Authors

Dear Author

Your manuscript is an interesting one but the data can be presented in a table. A table that summarized all studies on cfDNA epigenetics in several cancers.

Response: We thank the reviewer for this thorough and constructive review of our manuscript. We also thank the reviewer very much for this helpful suggestion. We have created a table summarizing the key publications dealing with tissue deconvolution and tissue-of-origin analysis to give the reader a better overview on the available data. However, we have decided to only summarize key publications covering all epigenetic features to keep the table in a reasonable size.

Sincerely

Fatemeh

Reviewer 3 Report

Oberhofer A. et al. present a review manuscript  dealing with various epigenetic features of cfDNA that can be used to deconvolute the data into cell-/tissue-specific signal.

Several other reviews have been written about this, but since the field is rapidly moving, updates are always welcome.

The manuscript is well-organized, as it is divided in sections based on the each epigenetic feature presented.

I have a few suggestions for improvement:

  • Figure 1, panel d: it might be misleading as it is similar to well-known t-SNE/UMAP plots in scRNAseq data. I suggest a more conservative figure such as a barplot with percentages of reads for each tissue type, or genomes equivalent.
  • Each section should be reorganized as follows: epigenetic mark; techniques used to study the mark (general); for each technique: specific cfDNA adaptation (if any); tools used to process the data; example of application with results.
  • Section 2.1: there are additional methods used for DNA methylation detection that should be described (see doi: 10.3390/biom10121677 for a recent review). Moreover, it should be made clear that the approaches described for DNA methylation detection (lines 117-138) are just general approaches, and their application to cfDNA requires specific modifications to the protocol. Some references to these “cfDNA-adapted protocols” are present, but some are missing (listing both): for microarray (Moss et al, Nat Comm 2018); for targeted BS-seq (M.C. Liu, Annals Oncol 2020); cfRRBS (A. De Koker, biorxiv 2019 - R. Van Paemel, Epigenetics 2021); cfMeDIP (D. De Carvalho group - Nat Protocols 2019); cfDNA TAPS (P. Siejka-ZieliÅ„ska, Sci Advances 2021); cfNOME (Erger, Genome Med 2020).
  • Fragmentomics (line 449 onward): the newly discovered population of ultra-short single-stranded cfDNA is not mentioned ( https://doi.org/10.1016/j.isci.2022.104554; doi.org/10.1101/gr.275691.121; doi.org/10.1186/s12915-021-01160-8 ). 
  • Section 2.6: Bioinformatic Analysis. Since each epigenetic feature has different specifics, I would include the bioinformatic approach to each epigenetic feature’s section. You can generate a Table summarizing each tool/approach in this section. I would describe in this section only the approaches that combine multiple features together.
  • A "Future Perspectives" section or paragraph(s) is desirable.

Author Response

Reviewer 3

Comments and Suggestions for Authors

Oberhofer A. et al. present a review manuscript  dealing with various epigenetic features of cfDNA that can be used to deconvolute the data into cell-/tissue-specific signal.

Several other reviews have been written about this, but since the field is rapidly moving, updates are always welcome.

The manuscript is well-organized, as it is divided in sections based on the each epigenetic feature presented.

Response: We thank our reviewer for this thorough and constructive review of our manuscript. We have rearranged and extended the manuscript intensively and hope to meet the reviewer’s expectation with those modifications.

I have a few suggestions for improvement:

  • Figure 1, panel d: it might be misleading as it is similar to well-known t-SNE/UMAP plots in scRNAseq data. I suggest a more conservative figure such as a barplot with percentages of reads for each tissue type, or genomes equivalent.

Response: We thought intensively about how to present the tissue deconvolution part and opted for the current depiction for two main reasons. First, we think that the underlying bioinformatic classification (i.e. the tissue deconvolution) of contributing cell/tissue types is best represented by this scheme. Second, several other publications (e.g., Heitzer et al. Nat Genet. Rev. 2018) used this plot for tissue deconvolution and without the axis labels and in this context, we are certain that the reader will be able to differentiate it from t-SNE/UMAP plots in scRNA-seq data.

  • Each section should be reorganized as follows: epigenetic mark; techniques used to study the mark (general); for each technique: specific cfDNA adaptation (if any); tools used to process the data; example of application with results.

Response: We reorganized the sections as suggested by the reviewer and included cfDNA adapted protocols wherever available.

  • Section 2.1: there are additional methods used for DNA methylation detection that should be described (see doi: 10.3390/biom10121677 for a recent review). Moreover, it should be made clear that the approaches described for DNA methylation detection (lines 117-138) are just general approaches, and their application to cfDNA requires specific modifications to the protocol. Some references to these “cfDNA-adapted protocols” are present, but some are missing (listing both): for microarray (Moss et al, Nat Comm 2018); for targeted BS-seq (M.C. Liu, Annals Oncol 2020); cfRRBS (A. De Koker, biorxiv 2019 - R. Van Paemel, Epigenetics 2021); cfMeDIP (D. De Carvalho group - Nat Protocols 2019); cfDNA TAPS (P. Siejka-ZieliÅ„ska, Sci Advances 2021); cfNOME (Erger, Genome Med 2020).

Response: We thank the reviewer very much for these detailed suggestions. We have restructured the paragraph dealing with techniques for DNA methylation profiling and mentioned several additional important detection methods for which cfDNA-adapted protocols exist. For a comprehensive and detailed review on DNA methylation profiling, we also refer to the reviews suggested by the reviewer.

  • Fragmentomics (line 449 onward): the newly discovered population of ultra-short single-stranded cfDNA is not mentioned ( https://doi.org/10.1016/j.isci.2022.104554; doi.org/10.1101/gr.275691.121; doi.org/10.1186/s12915-021-01160-8 ). 

Response: We have added a few sentences mentioning the newly discovered class of ultrashort single-stranded cfDNA and pointed out that this novel cfDNA population might also be interesting for tissue-of-origin analysis.

  • Section 2.6: Bioinformatic Analysis. Since each epigenetic feature has different specifics, I would include the bioinformatic approach to each epigenetic feature’s section. You can generate a Table summarizing each tool/approach in this section. I would describe in this section only the approaches that combine multiple features together.

Response: We have extended the description of bioinformatic analysis within the individual sections. Additionally, we have included the bioinformatic tool/approach within the table summarizing the key tissue-of-origin analyses that was suggested by the other two reviewers. We still opted for keeping the section on bioinformatic tools for general remarks on performing a sound bioinformatic analysis and to mention examples of approaches that combine multiple features together.

  • A "Future Perspectives" section or paragraph(s) is desirable.

Response: We have added a few paragraphs on Future Perspectives to the Conclusions section.

Round 2

Reviewer 3 Report

The authors have adequately addressed my comments.